# Tuning the reactivity of alkoxyl radicals from 1,5-hydrogen atom transfer to 1,2-silyl transfer

Zhaoliang Yang[1,4], Yunhong Niu[1,4], Xiaoqian He[2], Suo Chen[1], Shanshan Liu [1], Zhengyu Li[1], Xiang Chen[1], Yunxiao Zhang[1], Yu Lan [2,3✉] & Xiao Shen [1✉]

Controlling the reactivity of reactive intermediates is essential to achieve selective transformations. Due to the facile 1,5-hydrogen atom transfer (HAT), alkoxyl radicals have been proven to be important synthetic intermediates for the δ-functionalization of alcohols. Herein, we disclose a strategy to inhibit 1,5-HAT by introducing a silyl group into the α-position of alkoxyl radicals. The efficient radical 1,2-silyl transfer (SiT) allows us to make various α-functionalized products from alcohol substrates. Compared with the direct generation of α-carbon radicals from oxidation of α-C-H bond of alcohols, the 1,2-SiT strategy distinguishes itself by the generation of alkoxyl radicals, the tolerance of many functional groups, such as intramolecular hydroxyl groups and C-H bonds next to oxygen atoms, and the use of silyl alcohols as limiting reagents.

[1] The Institute for Advanced Studies, Engineering Research Center of Organosilicon Compounds and Materials, Ministry of Education, Wuhan University, Wuhan, People's Republic of China. [2] School of Chemistry and Chemical Engineering, Chongqing Key Laboratory of Theoretical and Computational Chemistry, Chongqing University, Chongqing, People's Republic of China. [3] College of Chemistry and Molecular Engineering, Zhengzhou University, Zhengzhou, People's Republic of China. [4] These authors contributed equally: Zhaoliang Yang, Yunhong Niu. ✉email: lanyu@cqu.edu.cn; xiaoshen@whu.edu.cn

adicals, anions, cations, carbenes, and others are key reactive intermediates in synthesis[1]. These intermediates usually show different reactivity, facilitating the development of complementary methodologies for the synthesis of molecules that are important in material and/or life-related field[2–4]. Among various radicals, alkoxyl radicals have gained increasing attention (Fig. 1a)[5–9]. Although the previous studies to generate alkoxyl radicals usually need pre-activated alcohols or corresponding precursors[10–21], recent work on the direct activation of alcohols with photocatalysis and/or transition-metal catalysis largely broaden their synthetic utility[22–29]. When δ-C–H bonds are present, the intramolecular 1,5-hydrogen atom transfer (HAT) from the δ-position via a low-energy six-membered ring transition state is usually favored over the transfer of hydrogen atoms at other positions, thus alkoxyl radical-mediated δ-C–H functionalization is widely studied (Fig. 1b)[6]. For example, the synthesis of δ-alkoxylimino alcohols and intermolecular δ-heteroarylation of alcohols through 1,5-HAT of alkoxyl radicals have been achieved (Fig. 1b)[27–29]. However, alkoxyl radical-mediated α-functionalization of alcohols have not been reported[30,31]. It is also known that excess amount of alcohols are usually required in oxidative C–H functionalization reactions, and it is challenging to control the selectivity when multiple oxidizable C–H bonds are present in the substrate[30,31]. Silicon possesses empty *d* orbitals and C–Si bond is longer than C–H bond. We envisioned that 1,2-silyl transfer (SiT) of alkoxyl radical via three-membered ring transition state (also known as radical Brook rearrangement, RBR) might be easier than the corresponding 1,2-HAT and might be favored over 1,5-HAT process (Fig. 1c).

RBR was initially proposed to explain the cyclopropanation product of the photoreaction between acylsilanes and electron-poor olefins in 1981[32–35]. However, the synthetic application of RBR was nearly ignored in the following decades[35–38]. Until 2017, Smith and group found that benzylic radicals can be generated from the oxidation of hypervalent silicate intermediate by a photo-excited Ir complex[37]. In 2020, our group revealed that the Mn-catalyzed RBR is superior to anion Brook rearrangement in the direct trifluoroethanol transfer reactions[38]. Herein, we show that radical 1,2-SiT is favored over 1,5-HAT under Ag-catalysis conditions, and selective α-C–C bond formation reactions are achieved without any δ-functionalization product, in which the use of alcohols as limiting reagents in the reaction of oxime ethers

and the tolerance of various C–H bonds and benzyl alcohols demonstrate the synthetic potential of our methodology (Fig. 1c)[30,31].

## Results

### Radical 1,2-SiT in the synthesis of α-hydroxyl oxime ethers.
Oximes and oxime ethers are important synthetic building blocks, and they have also been found to be core structural motifs of multiple bioactive molecules[28,39–41]. In 2018, Jiao and co-workers reported the synthesis of δ-alkoxylimino alcohols through 1,5-HAT of alkoxyl radicals, but no α-alkoxylimino alcohols were isolated[28]. Previous methods to prepare α-hydroxyl oxime ethers mainly rely on the reduction of alkoxyliminyl substituted ketones, which themselves need multistep synthesis[41]. To the best of our knowledge, there is no report on radical-mediated synthesis of α-alkoxylimino alcohols. Therefore, we choose to investigate the reaction between α-silyl alcohol **1a** and sulfonyl oxime ether **2** to check whether α-functionalization product or δ-functionalization product can be obtained.

### Optimization of the reaction conditions for the synthesis of α-hydroxyl oxime ethers.
Previously, we found the Mn(II)/Mn(III)-catalyzed metal alkoxide (M-OR) homo-cleavage strategy was an efficient way to generate alkoxyl radicals for the direct transfer of trifluoroethanol and difluoroethanol units[38]. Therefore, we focused on the investigation of various transition-metal salts for M-OR homo-cleavage. After extensive investigations, we found that AgNO$_3$ was a better pre-catalyst than CuCl$_2$, FeCl$_3$, NiBr$_2$, Mn(OAc)$_3$, and AgI (Fig. 2, entries 1–6). When the reaction was carried out in MeCN/H$_2$O (v/v = 1:1) at 80 °C for 12 h with K$_2$S$_2$O$_8$ as an oxidant, a yield of 47% was afforded for compound **3a**, without any detection of δ-functionalization product (entry 6). When the solvent was changed to acetone/H$_2$O (v/v = 1:1), the yield of α-functionalization product **3a** increased to 51% (entry 7). Increasing the concentration of the reaction resulted in an improved yield of compound **3a** (71%, entry 8). Other oxidants such as Na$_2$S$_2$O$_8$, (NH$_4$)$_2$S$_2$O$_8$, Dess–Martin periodinane, and *tert*-butyl peroxybenzoate afforded lower efficiency of the reaction (entries 9–12). Lowering the reaction temperature also resulted in a decreased yield of compound **3a** (entries 13 and 14). The control experiment showed that, without AgNO$_3$, only 20% yield of compound **3a** was observed by proton

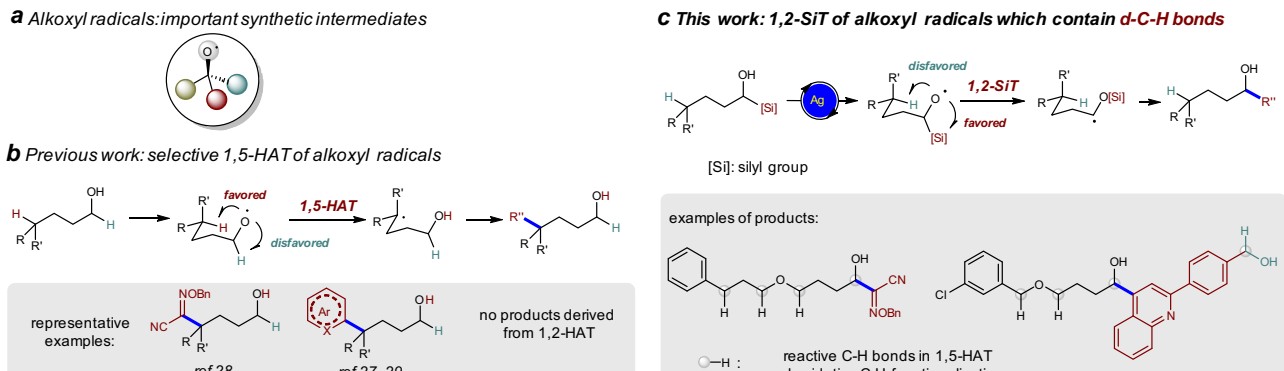

**a** *Alkoxyl radicals: important synthetic intermediates*

**b** *Previous work: selective 1,5-HAT of alkoxyl radicals*

representative examples:

ref 28   ref 27, 29   no products derived from 1,2-HAT

**c** *This work: 1,2-SiT of alkoxyl radicals which contain d-C-H bonds*

[Si]: silyl group

examples of products:

○—H : reactive C-H bonds in 1,5-HAT and oxidative C-H functionalizations

**Fig. 1 Tuning the reactivity of alkoxyl radicals from 1,5-HAT to 1,2-SiT by the incorporation of a silyl group. a** Alkoxyl radicals are important synthetic intermediates. **b** Intramolecular 1,5-hydrogen atom transfer (HAT) via a six-membered ring transition state is usually favored over the transfer of hydrogen atoms at other positions, thus alkoxyl radicals mediated δ-C–H functionalization is widely studied. **c** We disclose here that 1,2-silyl transfer (SiT) is favored over 1,5-HAT under Ag-catalyzed conditions, allowing the efficient synthesis of α-hydroxyl oxime ethers and α-heteroaryl alcohols (this work). The alkoxyl radical-mediated reactions can tolerate many reactive C–H bonds, which are reactive in 1,5-HAT and oxidative C–H functionalizations. Another OH group in the substrates can also be tolerated.

**Fig. 2 Reaction optimization.** [a]The mixture of **1a** (0.2 mmol), catalyst (0.04 mmol), oxidant (0.4 mmol), and **2** (0.3 mmol) in the solvent (2 mL) was stirred at $T$ under $N_2$ for 12 h. Conversions of **1a** and **2** and yield of **3a** were determined by [1]H NMR using $BrCH_2CH_2Br$ as an internal standard. [b]Solvent (1.5 mL). DMP Dess–Martin periodinane, TBPB *tert*-butyl peroxybenzoate.

| Entry | Catalyst | Oxidant | Solvent (V/V=1:1) | T/°C | Conv. of **1a** (%) | Conv. of **2** (%) | Yield of **3a** (%) |
|---|---|---|---|---|---|---|---|
| 1 | $CuCl_2$ | $K_2S_2O_8$ | $MeCN:H_2O$ | 80 | 94 | 94 | 15 |
| 2 | $FeCl_3$ | $K_2S_2O_8$ | $MeCN:H_2O$ | 80 | 95 | 44 | 20 |
| 3 | $NiBr_2$ | $K_2S_2O_8$ | $MeCN:H_2O$ | 80 | >99 | 50 | 29 |
| 4 | $Mn(OAc)_3$ | $K_2S_2O_8$ | $MeCN:H_2O$ | 80 | >99 | 93 | 33 |
| 5 | AgI | $K_2S_2O_8$ | $MeCN:H_2O$ | 80 | 96 | 90 | 14 |
| 6 | $AgNO_3$ | $K_2S_2O_8$ | $MeCN:H_2O$ | 80 | >99 | 61 | 47 |
| 7 | $AgNO_3$ | $K_2S_2O_8$ | $Acetone:H_2O$ | 80 | >99 | 96 | 51 |
| 8[b] | $AgNO_3$ | $K_2S_2O_8$ | $Acetone:H_2O$ | 80 | >99 | 86 | 71 |
| 9[b] | $AgNO_3$ | $Na_2S_2O_8$ | $Acetone:H_2O$ | 80 | >99 | 96 | 50 |
| 10[b] | $AgNO_3$ | $(NH_4)_2S_2O_8$ | $Acetone:H_2O$ | 80 | >99 | 93 | 55 |
| 11[b] | $AgNO_3$ | DMP | $Acetone:H_2O$ | 80 | >99 | 26 | <1 |
| 12[b] | $AgNO_3$ | TBPB | $Acetone:H_2O$ | 80 | 28 | <2 | <1 |
| 13[b] | $AgNO_3$ | $K_2S_2O_8$ | $Acetone:H_2O$ | 50 | >99 | 94 | 55 |
| 14[b] | $AgNO_3$ | $K_2S_2O_8$ | $Acetone:H_2O$ | 25 | 72 | 53 | 32 |
| 15[b] | / | $K_2S_2O_8$ | $Acetone:H_2O$ | 80 | 78 | 91 | 20 |
| 16[b] | $AgNO_3$ | / | $Acetone:H_2O$ | 80 | 4 | 3 | <1 |

1) with **1a** ([Si] = $PhMe_2Si$), >99 conv. of **1a**, 86% conv. of **2**, **71% yield of 3a**;
2) with **1ab** ([Si] = $Me_3Si$), >99 conv. of **1ab**, 91% conv. of **2**, **41% yield of 3a**;
3) with **1ac** ([Si] = $t$-$BuMe_2Si$), >99 conv. of **1ac**, 92% conv. of **2**, **13% yield of 3a**;
4) with **1ad** ([Si] = $Ph_2MeSi$), 92% conv. of **1ad**, 90% conv. of **2**, **48% yield of 3a**;
5) with **1ae** ([Si] = $Ph_3Si$), 81% conv. of **1ae**, 72% conv. of **2**, **31% yield of 3a**.

**Fig. 3 Influence of silyl groups on the efficiency of the reaction.** Reaction conditions: the mixture of **1** (0.2 mmol), $AgNO_3$ (0.04 mmol), $K_2S_2O_8$ (0.4 mmol), **2** (0.3 mmol), and acetone/$H_2O$ (0.75 mL/0.75 mL) was stirred at 80 °C under $N_2$ for 12 h. Conversion of **1** and **2** and yield of **3a** were determined by [1]H NMR using $BrCH_2CH_2Br$ as an internal standard.

nuclear magnetic resonance, although a large amount of decomposition of compounds **1a** and **2** (entry 15) were found. However, no **3a** was generated without $K_2S_2O_8$, and the conversions of compounds **1a** and **2** were also low, indicating that $AgNO_3$ alone cannot catalyze the reaction (entry 16).

**Influence of silyl groups on the efficiency of the reaction.** Encouraged by the favored α-functionalization over δ-functionalization in the reaction between compounds **1a** and **2**, we then investigated the influence of the silyl substituent on the efficiency of the desired α-functionalization reaction. It was found that both electronic property and steric hindrance of the silyl

group showed a significant effect on our reaction. The electron-withdrawing effect of the phenyl group on the silicon atom appears to play a positive role in this reaction (Fig. 3). However, the steric hindrance on the silicon atom shows a negative effect in this reaction (**1a** vs **1ad** and **1ae**; **1ab** vs **1ac**). In all cases, aldehyde derived from compound **1** was formed as by-product. The substituents might not only affect the transfer ability of the silyl group but also affect the stability and reactivity of the radical intermediate **III** (see below). Moreover, the different substituents of the silyl groups also affect the C–Si bond length and bond dissociation energy, which might also be important factors in 1,2-SiT. Again, none of the reactions afforded δ-functionalization product.

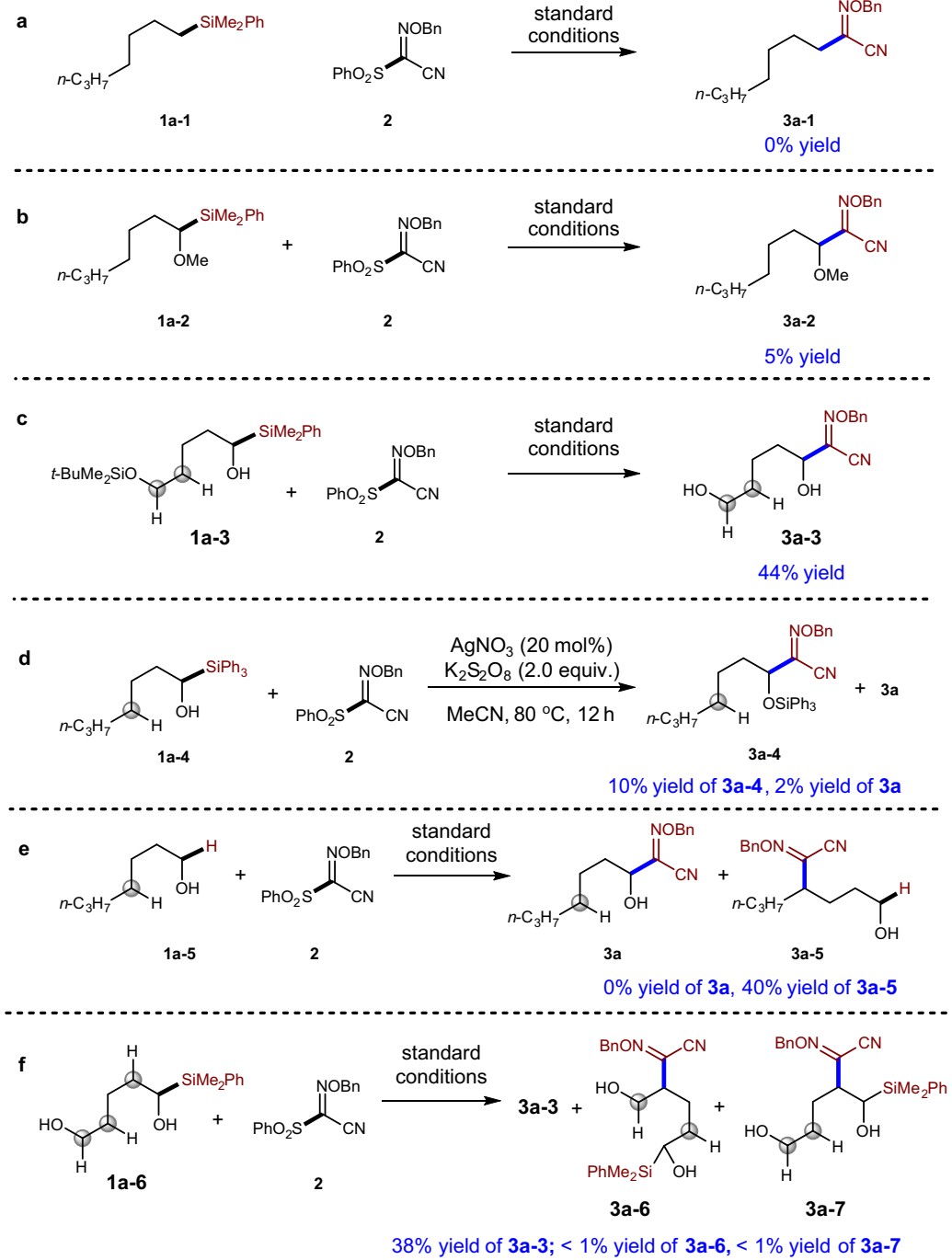

**Fig. 4 Mechanistic studies. a** Failure of the reaction with heptyldimethyl(phenyl)silane **1a-1** indictates the importance of OH group in the success of the reactions. **b** Failure of the reaction of protected α-silyl alcohol **1a-2** indicates that direct oxidative cleavage of C–Si bond is less likely. **c** Reaction **1a-3** shows that silyl ether group could be hydrolyzed under the reaction conditions. **d** Reaction of α-triphenylsilyl alcohol **1a-4** under conditions without H$_2$O could afford non-desilylated product **3a-4**. **e** The reaction of non-silylated alcohol **1a-5** indeed afforded 1,5-HAT product **3a-5** without the formation of α-functionalization product **3a**. **f** Remote OH group in **1a-6** can be tolerated.

**Mechanism study**. After identification of a suitable silyl group to promote the efficient synthesis of α-alkoxylimino alcohol **3a**, we set to investigate whether the reaction proceeded through radical 1,2-SiT or not. Firstly, our study reveals that the OH group is important for the success of the reaction. The use of compound **1a-1** as starting material resulted in no anticipated product **3a-1** (Fig. 4a). Protected α-silyl alcohol **1a-2** only gave 5% yield of

compound **3a-2** (Fig. 4b), indicating that the generation of carbon radical via direct oxidative cleavage of C–Si bond is less likely to be the major pathway in the reaction with **1a**[42–45]. This result was consistent to the similar oxidation potential of α-silyl alcohol and the methyl-protected counterpart[46]. When silyl ether compound **1a-3** was applied in the reaction with compound **2**, free diol **3a-3** was obtained in 44% yield (Fig. 4c), suggesting that silyl ether can

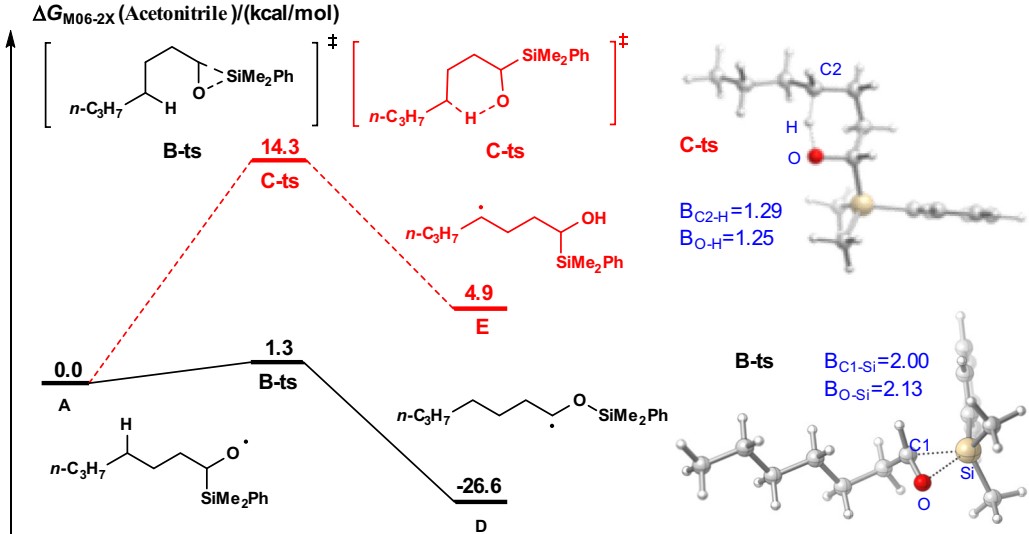

**Fig. 5 Free-energy profile of the pathways for 1,2-SiT vs 1,5-HAT.** The energies are in kcal/mol and represent the relative free energies calculated with the DFT/M06-2X method in MeCN. The bond distances are in angstroms.

be hydrolyzed under aqueous reaction condition. Further study of the reaction of triphenylsilyl-substituted alcohol **1a-4** with compound **2** under no $H_2O$ condition showed that compound **3a-4** could be synthesized in 10% yield with 2% yield of desilylated compound **3a** (Fig. 4d). The lower yield of **3a-4** and **3a** might be explained by the low solubility under the non-aqueous conditions (Fig. 4d). Jiao and co-workers have shown that alcohols can participate in δ-functionalization via radical 1,5-HAT under Ag catalysis[28]. The reaction of non-silylated alcohol **1a-5** indeed afforded 1,5-HAT product **3a-5** in 40% yield without the formation of α-functionalization product **3a** (Fig. 4e). Interestingly, when a silyl alcohol **1a-6**, which contains another C–OH bond, was tested in the reaction with compound **2**, the major product is C–Si bond functionalization product **3a-3** (38% yield; Fig. 4f), further indicating radical 1,2-SiT is favored over 1,5-HAT. The tolerance of free alcohol is an advantage of our method, since diols are challenging substrates for the oxidative C–H bond functionalization chemistry[30,31].

Subsequently, we investigated the energy barrier of 1,2-SiT and 1,5-HAT of alkoxyl radical intermediate **A** using density functional theory (DFT) calculation employing the method M06-2X (for details, see the Supplementary information 3i–n)[47–49]. As shown in Fig. 5, alkoxyl radical **A** was set as the starting point for the free-energy profiles. 1,2-SiT via transition state **B-ts** to generate radical intermediate **D** is quite easy with an energy barrier of only 1.3 kcal/mol, and this process is exothermic (26.6 kcal/mol). However, 1,5-HAT via transition state C-ts to generate radical intermediate **E** is an endothermic reaction (4.9 kcal/mol) with an energy barrier of 14.3 kcal/mol. The calculation results show that 1,2-SiT process of radical **A** is both dynamically and thermodynamically favored over the corresponding 1,5-HAT.

Based on our experimental DFT calculation results and previous reports[28], a plausible mechanism involving 1,2-SiT was proposed in Fig. 6. Oxidation of Ag(I) by $K_2S_2O_8$ might afford Ag (II)[50], which would undergo ligand exchange with alcohol **1a** and results in the generation of intermediate **I**. Homolysis of intermediate **I** could produce alkoxyl radical **II** and Ag(I). Carbon radical **III** would be generated through 1,2-SiT, which is favored over 1,5-HAT. Intermediate **IV** might be generated from the addition–elimination process between carbon radical **III** and compound **2**, but we cannot rule out the possibility of its

formation through trapping **III** with iminyl radical generated from homolysis of compound **2** (for details, see Supplementary information). $PhSO_2$ radical might be converted to $PhSO_3H$ under the oxidation condition in the aqueous solution[15]. $PhSO_3^-$ was detected by high-resolution mass spectrometry analysis of the reaction mixture, which supports this proposal (for details, see Supplementary information). The alcohol product **3** would be produced after desilylation under aqueous condition.

**Scope of the reaction between α-silyl alcohol** 1a **and sulfonyl oxime ether** 2. Subsequently, we investigated the scope of the radical substitution reaction between α-silyl alcohol **1** and sulfonyl oxime ether **2**. The reaction showed broad substrate scope, and various α-silyl alcohols could participate in the reaction, affording corresponding α-alkoxylimino alcohols in 48–70% yields. When 1 g of **1a** was employed, product **3a** could be isolated in 60% yield. The reaction can tolerate many functional groups, such as $C(sp^3)$-Br, $C(sp^3)$-$N_3$, $C(sp^2)$-F, $C(sp^2)$-Cl, $C(sp^2)$-Br, $C(sp^2)$-CN, $C(sp^2)$-$OCF_3$, and an ester group. These functional groups can be used for further transformations. When the δ-C–H bond is next to an oxygen atom, the 1,5-HAT of the alkoxyl radicals could be more favored, because the new radical can be stabilized by the oxygen through hyper-conjugation interaction[30,31]. However, under our reaction conditions, not only the normal δ-C–H bond can be tolerated, but the δ-C–H bond next to an oxygen atom can also be tolerated (Fig. 7, **3g**, **3i–3o**, **3s**). Moreover, benzylic, α-oxy, and α-benzoyloxy C–H groups, which are usually reactive in oxidative C–H bond cleavage reactions, are maintained under our reaction conditions (Fig. 7, **3g–3x**)[30,31]. In addition, our reaction can be applied in the synthesis of alcohols, which contain a β-substituent. Compound **3y** was synthesized in 65% yield, which is in sharp contrast to the failure to synthesize alcohols in previous Ag-catalyzed reaction[28]. The relative lower yield was found for the assembly of tertiary alcohol **3z** (20% yield), **3aa** (31% yield), and **3ab** (41% yield). In all cases, the alcohols **3** were obtained directly after the reaction, without the extra deprotection step of anticipated silyl ether intermediate **IV**. In all cases, the alcohol substrates were used as limiting reagents, which further highlight the synthetic potential of the current method since the oxidative α-C–H functionalization of alcohols usually need excess amount of alcohols, and in

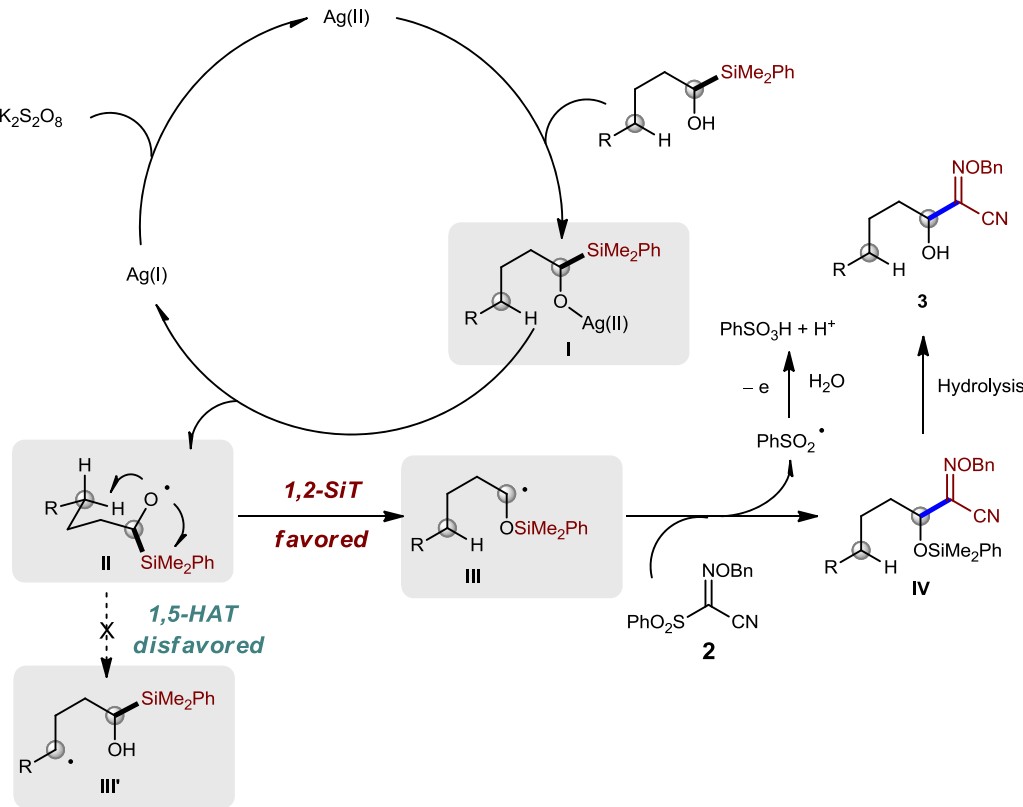

**Fig. 6 Proposed mechanism.** The proposed mechanism involves an Ag(I)/Ag(II) catalytic cycle and 1,2-SiT.

many cases, alcohols must be used as a solvent to achieve synthetic useful yield[30,31].

**Synthetic transformations of compound 3a.** Compound **3a** was easily transformed to methylated product **1a-2** in 57% yield, while the imine group was untouched (Fig. 8). The CN group could be hydrolyzed in the presence of $H_2O_2$ and $K_2CO_3$, affording amide **3a-8** in 72% yield (Fig. 8).

**Radical 1,2-SiT in the catalytic Minisci reaction for the synthesis of α-heteroaryl alcohols.** Heteroaryl groups are important structural motifs and Minisci reaction is one of the most atom-economic ways to introduce heteroaryl groups into organic molecules, by cleaving the C(sp2)–H bond[51,52]. The 1,5-HAT process of alkoxyl radicals was used by Zhu's group and Chen's group in the hypervalent iodine-mediated radical Minisci reaction and various δ-heteroaryl-substituted alcohols have been made[27,29]. To the best of our knowledge, there has been no report on Minisci type α-heteroarylation through 1,2-HAT of alkoxyl radicals. Although direct radical α-heteroarylation of alcohols was achieved by the intermolecular H abstraction, they usually need excess amount of alcohols as reagents[31,51,52]. In most cases, alcohols are used as the solvent, which limits the application of those methods, especially when complex alcohols are needed and/or the alcohols are solids. Encouraged by the success of the application of radical 1,2-SiT in the direct synthesis of alkoxylimino alcohols, we probed the applicability of this strategy in the catalytic Minisci reaction for the synthesis of secondary alkyl heteroaryl alcohols.

After a quick optimization of reaction conditions (for details, see Supplementary information), we found that similar Ag-catalysis conditions could be applied in the reactions between α-silyl alcohol **1** and heteroarenes **4** (Fig. 9). Quinolines with methyl and aryl

substituents are competent reaction partners, delivering the desired α-heteroarylation products **5a–k** in 53–79% yields. The F, Cl, Br, CN, OMe, and Me groups on the aryl substituents are tolerated. Isoquinoline derivatives such as Cl-, Br-, MeO-, and BnO-substituted isoquinolines performed well, affording products **5m–r** in 53–74% yields. An α-silyl alcohol containing a long alkyl chain also works, affording compound **5r** in 64% yield and **5t** in 60% yield. Moreover, phenanthridine can participate in the reaction, giving corresponding alkyl heteroaryl alcohol **5s** in 51% yield. One of the disadvantages of the previous direct α-heteroarylation of alcohol is the need to use excess amount of alcohol, which is formidable when the complex is applied. We found that only two equivalent of α-silyl alcohol was required in the current radical Minisci reaction, and the relatively complex alcohols **5v–z** were prepared in 50–62% yields. It is worth noting that even benzyl alcohol can be tolerated (**5aa**, 50%; **5ab**, 51%). These two compounds would be challenging to synthesize via the oxidative C–H bond functionalization methodology because the benzyl alcohol would result in trouble[30,31]. Moreover, the synthesis of alcohols, which contain a β-substituent, was successful and compound **5ac** was synthesized in 62% yield, which is in sharp contrast to the failure to synthesize alcohols in previous Ag-catalyzed reaction[28]. Again, no δ-heteroarylation products were isolated in all cases shown in Fig. 9.

**Discussion**

In this work, we found that the introduction of a silyl group to the α-position of alcohols can effectively inhibit 1,5-HAT of the corresponding alkoxyl radicals. The substituents on the silicon are found to be important to achieve efficient 1,2-SiT. The carbon radicals derived from 1,2-SiT are applied in the radical substitution reactions of sulfonyl oxime ether and heteroarenes to prepare α-alkoxylimino alcohols and alkyl heteroaryl alcohols.

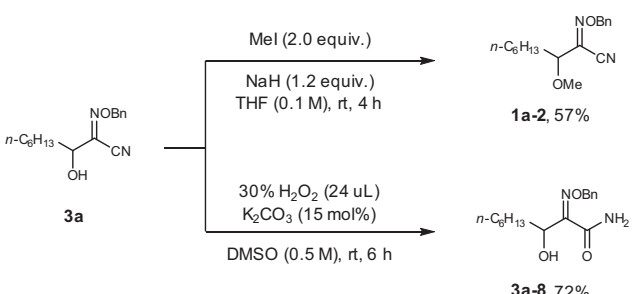

**Fig. 7 Scope for the synthesis of α-alkoxylimino alcohols.** [a]The yield in the parentheses refers to the isolated yield of the gram-scale reaction. [b]Acetone/H$_2$O (v/v = 2:1) was used as the solvent.

**Fig. 8 The chemo-selective transformation of 3a.** The oxime ether product **3a** could be easily converted to methylated product **1a-2** and amide **3a-8**.

Compared with the direct generation of α-carbon radicals from the oxidation of α-C–H bond of alcohols, the 1,2-SiT strategy distinguished itself by the generation of alkoxyl radicals, the tolerance of many functional groups such as intramolecular hydroxyl groups and C–H bonds next to oxygen atoms, and the use of silyl alcohols as limiting reagents. Our experimental finding broadens the synthetic application of alkoxyl radicals. Further application of the 1,2-SiT of alkoxyl radicals is underway in our laboratory.

## Methods

**Typical procedure 1 (3a).** In an Ar-protected glove box, AgNO$_3$ (6.8 mg, 0.04 mmol, 20 mol%), **2** (90.0 mg, 0.30 mmol, 1.5 equiv.), and **1a** (50.0 mg, 0.20 mmol) were added into a reaction tube. After that, the tube was taken out of the box, acetone/H$_2$O (0.75 mL/0.75 mL) and K$_2$S$_2$O$_8$ (108.0 mg, 0.40 mmol, 2.0 equiv.) were added under N$_2$. The tube was then sealed, and the resulting mixture was kept stirring at 80 °C in a heating block for 12 h. The reaction mixture was quenched with water (5 mL), extracted with ethyl acetate (3 × 10 mL), and the organic phase was combined and washed with brine, dried over anhydrous Na$_2$SO$_4$, and concentrated under reduced pressure. The crude product was purified with column chromatography on silica gel (200–300 mesh) with petroleum ether/ethyl acetate (PE/EA) (8/1, v/v) as eluent to afford 38.0 mg of the title compound as a faint yellow oil (69% yield).

**Typical procedure 2 (5a).** Under N$_2$ atmosphere, AgNO$_3$ (6.8 mg, 0.04 mmol, 20 mol%), CH$_3$CN/H$_2$O (1.67 mL/0.33 mL), **4a** (28.6 mg, 0.20 mmol), CF$_3$COOH (22.8 mg, 0.2 mmol, 1.0 equiv.), **1a** (100 mg, 0.40 mmol, 2.0 equiv.), and K$_2$S$_2$O$_8$ (118.8 mg, 0.44 mmol, 2.2 equiv.) were added into a reaction tube. The tube was then sealed, and the resulting mixture was kept stirring at 80 °C in a heating block for 12 h. The reaction mixture was quenched with saturated NaHCO$_3$

**Fig. 9 Scope of Minisci reaction for the synthesis of α-heteroaryl alcohols.** The reaction could tolerate various C–H bonds and benzyl alcohols that are reactive in oxidative C–H functionalization reactions. [a]DMSO/H$_2$O (v/v = 5:3) was used as the solvent.

aqueous solution (10 mL), extracted with ethyl acetate (3 × 10 mL), and the organic phase was combined and washed with brine, dried over anhydrous Na$_2$SO$_4$, and concentrated under reduced pressure. The crude product was purified with column chromatography on silica gel (200–300 mesh) with PE/EA (10/1, v/v) as eluent to afford 27.0 mg of the title compound as a faint yellow oil (53% yield).

## Data availability
The authors declare that all other data supporting the findings of this study are available within the article and Supplementary information files, and also are available from the corresponding author on reasonable request.

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

## Acknowledgements

We are grateful to NSFC (21901191, 21822303), Fundamental Research Funds for the Central Universities (2042018kf0023, 2042019kf0006), State Key Laboratory of Bioorganic & Natural Products Chemistry (BNPC18237), and Wuhan University for financial support. We are thankful to Prof. Aiwen Lei and Prof. Xumu Zhang at Wuhan University for the generous provision of the basic instruments.

## Author contributions

X.S. designed and directed the investigations and composed the manuscript with revisions provided by the other authors. Z.Y. and Y.N. developed the catalytic method. Z.Y., Y.N., and S.C. studied the substrate scope. X.H. and Y.L. conducted the calculations. Z.Y., Y.N., X.H., S.C., S.L., Z.L., X.C., Y.Z., Y.L., and X.S. were involved in the analysis of results and discussions of the project.

## Competing interests

The authors declare no competing interests.
