## [Peer Review File · Nature Communications]

REVIEWER COMMENTS

Reviewer #1 (Remarks to the Author):

Xiao Shen and coworkers described an α functionalization of alcohols from α -silyl alcohols. The featured part of this work is the α selectivity due to the favorable 1,2 silyl transfer than 1,5 HAT while generating alkoxy radical. The α radical could be trapped by both sulfonyl oxime ether and quinolines to afford a variety of α functionalized alcohols. The control experiments also indicate that the α silyl group played an important role to facilitate oxidation and following 1,2 SAT.

However, the reaction mechanism is not new, the way to generate alkoxy radical, 1,2 SAT and radical trap steps are all well studied. And the regioselectivity is not very impressive when you need to preinstall a functional group at the α position to get α selectivity.

Besides there are some questions need to be addressed:

1. To compare with previous direct α C-H functionalization of alcohols, is there a regioselective single step method to construct α silyl group onto alcohols efficiently? The starting material synthesis is not included in supporting information.
2. The reaction is not well optimized. Only moderate yield was achieved. In figure 6, most of substrates only have 50%-60% yield. And there are only two examples have 70% yield which is the highest yield in this figure.
3. For Minisci reaction, the substrate scope is limited in quinolines. How about medicinally more relevant pyridines?

Overall, it is my opinion that publication at this time would be premature.

Reviewer #2 (Remarks to the Author):

The authors disclose two practical advances that rely on a conceptual insight – the recognition that, once an alkoxy radical is generated, 1,2-silyl transfer would be more kinetically feasible than 1,5-hydrogen atom transfer (1,5-HAT), and the resultant α -alkoxy radical could be trapped by known trapping agents such as an oxime ether (c.f. Table 1, Figure 6), or a heteroarene (Figure 7). In the latter case, the manuscript may be technically correct when it states, "To the best of our knowledge, there have been no report on Minisci type α -heteroarylation through 1,2-HAT of alkoxy radicals" (Figure 7). The closest related precedents are quite close (references 31 and 37). In reference 31, Lei and co-workers disclose Minisci type α -heteroarylation reactions, which use a 20-fold excess of alcohol substrate, rather than the 2-fold excess used in the reactions disclosed in this Nature Commun. (Figure 7). In reference 37, Smith et al engage arylsilylmethanol (3 equiv) in radical-mediated α -(hetero)arylation reactions (1 equiv). In spite of this close proximity, Figure 7 of the submitted manuscript constitutes a practical and meaningful improvement over known technologies. When paired with this explicit conceptual recognition, this research could merit publication in Nature Commun. following revisions to address the below opportunities.

Control experiments are included that are consistent with the proposed reaction mechanism.

- Additional control experiments would be of benefit when interpreting experiments designed to probe reaction mechanisms. For example, an experiment involving TEMPO is used to support the claim that the developed reactions are radical mediated (Figure 3a). Unfortunately, as TEMPO is an oxidant, this experiment is impossible to interpret in the absence of three control reactions with TEMPO, and without each of the oxidants included in the showcased reaction (-AgNO₃, -K₂S₂O₈, or without either of these oxidants) to suggest that the reaction is not likely to change mechanism based on the inclusion of TEMPO in the reaction medium. Even with these control reactions, an

experiment involving TEMPO would likely prove inconclusive.

- Control reactions are consistent with reaction occurring at oxygen as a prerequisite to formation of the alkyl radical (Figure 3b-c).
- Control reactions are consistent with preferential 1,5-HAT from an alkoxy radical in the absence of an α -disposed silyl group (Figure 3f).

DFT calculations are included (Figure 4) that are consistent with a lower energy pathway for 1,2-silyl transfer relative to 1,5-HAT. The manuscript would be improved by inclusion of citations to support the choice of this basis set and functional, and explicit acknowledgement of the limitations of these DFT calculations (c.f. Chem. Sci. 2020, 11, 217 & references therein). The documentation for these DFT calculations in the supporting information is inadequate for their evaluation, and must be improved.

The proposed mechanism (Figure 5) is plausible, and citations explicitly related to the roles of oxidants need to be included in the context of the associated discussion either in the manuscript or supporting information.

With a cursory scan of Figures 6 and 7, the grey circles on top of some carbon atoms appear similar to oxygen atoms, so it may be worthwhile to reconsider whether their use in this document is beneficial.

Supporting information (SI):

1. Many procedures appear well drafted.
2. Many ^{13}C NMR spectra in the supporting information do not meet the ACS definition for characterization-quality spectra and do not provide adequate evidence that the compounds can be isolated in high purity (c.f. pages S54, S58, S62, S70, etc).

Reviewer #3 (Remarks to the Author):

In this manuscript, Yang, et. al. introduces a silyl group at α -position of alcohol that can inhibit 1,5-hydrogen atom transfer (1,5-HAT) and make various α -functionalization products from alcohol substrate. Notably, this method employed silver catalyst and use cheap oxidant. Author also bring a novelty to selectively transform alcohol through 1,2-silyl transfer (1,2-SiT) instead of 1,5-HAT. The significance of this paper for the targeted journal is good. This manuscript provides broad substrate scopes, several mechanistic studies and computational calculation to support the plausible mechanism. However, scopes of alcohol were limited to secondary alcohol. The manuscript may be publishable once the following items have been addressed. My detailed analysis of this manuscript are as follow

Specific comments

1. In influence of silyl groups on the efficiency of the reaction (Section 1.2), is the different substrate of silyl group affect C-Si bond length and BDE? I think C-Si bond and BDE is also important factor in 1,2SiT.
2. In mechanism study (Fig. 3c), "Protected α -silyl alcohol 1a-2 only gave 5% yield of compound 3a-2 (Fig. 3c), indicating that the generation of carbon radical via direct oxidative cleavage of C-Si bond is less likely to be the major pathway in the reaction with 1a." These results show that a direct oxidation mechanism of α -silyl alcohol is also possible. In general, when a silyl group is introduced at the α -position of an alcohol, the oxidation potential is reduced compared to the free alcohol. But α -silyl alcohol still has high oxidation potential Therefore, a strong oxidant is necessary. In order to exclude the direct oxidation mechanism, it is recommended to add information on the oxidation potential of alpha silyl alcohol and related reactions to the reference.

J. Am. Chem. Soc. 1990, 112, 1962-1970. Angew. Chem. Int. Ed. 1998, 37, 660-662. Org. Lett. 2017, 19, 4696-4699. Org. Lett. 2018, 20, 6239-6243. Chem. Commun. 2020, 56, 2873-2876.

3. In mechanistic study (Fig.3e), the role of water was examined with a reaction of standard condition without water as a solvent. In this case, why triphenyl silyl substituted alcohol 1a-4 used instead of phenyl dimethyl silyl substituted alcohol? Meanwhile another mechanistic study used phenyl dimethyl silyl substituted alcohol.

4. In alcohol scopes, there were many secondary alcohol scopes but just only one example of tertiary alcohol. Do tertiary alcohol substrates fail under the report reaction conditions or were they not tried? I think it would be of interest to the readership for this information to be provided.

5. In the Menisci type reaction, quinoline type of substrates were studied, the more general heterocycles weren't considered. It is recommended to add reactions of various heterocyclic compounds.

6. In this manuscript there were no gram scale reaction and further application (i.e. transformation further of product) to check the valuable of the product.

Minor Comments

1. Fig. 4 caption: "Free-energy profile of the pathways for 1,2-SiT vs 1,5-H AT..."
1,5-HAT sentences should not contain space between H and AT

2. In SI page S5, compound 1h and 3h are same. Please fix it.

3. In SI NMR spectra, the quality were good but some of spectra still contain impurity or solvent junk. I think the addition of title in each spectrum would be better (i.e. ¹H NMR spectrum, ¹³C NMR spectrum, or ¹⁹F NMR spectrum) and also insets with enlarged multiplets should be included on spectra.

Response to Referees

Reviewer #1 (Remarks to the Author):

Xiao Shen and coworkers described an alpha functionalization of alcohols from alpha-silyl alcohols. The featured part of this work is the alpha selectivity due to the favorable 1,2 silyl transfer than 1,5 HAT while generating alkoxy radical. The alpha radical could be trapped by both sulfonyl oxime ether and quinolines to afford a variety of alpha functionalized alcohols. The control experiments also indicate that the alpha silyl group played an important role to facilitate oxidation and following 1,2 SAT.

However, the reaction mechanism is not new, the way to generate alkoxy radical, 1,2 SAT and radical trap steps are all well studied. And the regioselectivity is not very impressive when you need to preinstall a functional group at the alpha position to get alpha selectivity.

Besides there are some questions need to be addressed:

1. To compare with previous direct alpha C-H functionalization of alcohols, is there a regioselective single step method to construct alpha silyl group onto alcohols efficiently? The starting material synthesis is not included in supporting information.

Response: Thanks for the comments. There are single step methods to construct alpha silyl group onto alcohols (*Angew. Chem. Int. Ed.* **2011**, *50*, 6375-6378; *Org. Lett.* **2013**, *15*, 1524-1527; *Synthesis*, 2004 786-790; *Tetrahedron Lett.* 2011, *52*, 422-425). But people do not necessarily need to use alcohol as the substrates to prepare alpha-silyl alcohols. alpha-Silyl alcohols could be prepared by nucleophilic silylation of aldehydes. The general procedures to synthesize alpha-silyl alcohols are provided in the supplementary information (Part 7 Synthesis of α -silyl alcohols).

2. The reaction is not well optimized. Only moderate yield was achieved. in figure 6, most of substrates only have 50%-60% yield. And there are only two examples have 70% yield which is the highest yield in this figure.

Response: Thanks for the comments. However, we insist that the yield are synthetically useful.

3. for Minisci reaction, the substrate scope is limited in quinolines. How about medicinally more relevant pyridines?

Response: Thanks for the comments. Quinolines and isoquinolines are suitable substrates. We tried many pyridines, such as **B1**, **B2**, **B3** and **B4**, but less than 5% yield were afforded (**Figure R-1**). Other heteroarenes such as **B5**, **B6**, **B7**, **B8** and **B9** were also not successful substrates under the current conditions. A yield of 9% were obtained when **B10** was employed. Further optimization of the reaction conditions with **B10** as the substrate resulted in 35% isolated yield of product **C10** (**Figure R-2**). We added the data of **C10** in the revised manuscript. The failed substrates were added in the revised supplementary information.

The yield was determined by ^1H NMR using mesitylene as an internal standard.

Figure R-1

Entry	Solvent	Yield. of C10
1	Acetone:H ₂ O (5:1)	5
2	DMSO:H ₂ O (5:1)	5
3	DMSO:H ₂ O (5:2)	19
4	DMSO:H ₂ O (5:3)	25(35 ^a)
5	DMSO:H ₂ O (5:4)	19
6	DMSO:H ₂ O (5:5)	8

Yield of **C10** were determined by ^1H NMR using Mesitylene as an internal standard. ^a isolated yield

Figure R-2

Overall, it is my opinion that publication at this time would be premature.

Response: We thank reviewer #1 for the comment. However, we insist that our work are both synthetically and conceptually important. Reviewer #2 commented: “The authors disclose two practical advances that rely on a conceptual insight – the recognition that, once an alkoxy radical is generated, 1,2-silyl transfer to would be more kinetically feasible than 1,5-hydrogen atom transfer (1,5-HAT)”. Reviewer #3 commented: “Figure 7 of the submitted manuscript constitutes a practical and meaningful

improvement over known technologies” and “explicit conceptual recognition”.

Reviewer #2 (Remarks to the Author):

The authors disclose two practical advances that rely on a conceptual insight – the recognition that, once an alkoxy radical is generated, 1,2-silyl transfer would be more kinetically feasible than 1,5-hydrogen atom transfer (1,5-HAT), and the resultant α -alkoxy radical could be trapped by known trapping agents such as an oxime ether (c.f. Table 1, Figure 6), or a heteroarene (Figure 7). In the latter case, the manuscript may be technically correct when it states, “To the best of our knowledge, there have been no reports on Minisci type α -heteroarylation through 1,2-HAT of alkoxy radicals” (Figure 7). The closest related precedents are quite close (references 31 and 37). In reference 31, Lei and co-workers disclose Minisci type α -heteroarylation reactions, which use a 20-fold excess of alcohol substrate, rather than the 2-fold excess used in the reactions disclosed in this Nature Commun. (Figure 7). In reference 37, Smith et al engage arylsilylmethanol (3 equiv) in radical-mediated α -(hetero)arylation reactions (1 equiv). In spite of this close proximity, Figure 7 of the submitted manuscript constitutes a practical and meaningful improvement over known technologies. When paired with this explicit conceptual recognition, this research could merit publication in Nature Commun. following revisions to address the below opportunities.

Response: Thanks for the positive comments.

Control experiments are included that are consistent with the proposed reaction mechanism.

- Additional control experiments would be of benefit when interpreting experiments designed to probe reaction mechanisms. For example, an experiment involving TEMPO is used to support the claim that the developed reactions are radical mediated (Figure 3a). Unfortunately, as TEMPO is an oxidant, this experiment is impossible to interpret in the absence of three control reactions with TEMPO, and without each of the oxidants included in the showcased reaction (-AgNO₃, -K₂S₂O₈, or without either of these oxidants) to suggest that the reaction is not likely to change mechanism based on the inclusion of TEMPO in the reaction medium. Even with these control reactions, an experiment involving TEMPO would likely prove inconclusive.
- Control reactions are consistent with reaction occurring at oxygen as a prerequisite to formation of the alkyl radical (Figure 3b-c).
- Control reactions are consistent with preferential 1,5-HAT from an alkoxy radical in the absence of an α -disposed silyl group (Figure 3f).

Response: Thanks for the suggestions. Three control experiments were performed (**Figure R-3**, entries 2-4). In the presence of TEMPO, without AgNO₃ and K₂S₂O₈, there were only 2% conversion of **1a** and **2**, and no formation of **3a** was detected (**Figure R-3**, entry 4), indicating that the addition of TEMPO is less likely to change the mechanism. We added these data to the revised supplementary information.

Entry	variation from standard conditions	conversion (1a)	conversion (2)	Yield (3a)
1	no variation	68%	17%	7%
2	without K ₂ S ₂ O ₈	7%	11%	0%
3	without AgNO ₃	48%	20%	13%
4	without K ₂ S ₂ O ₈ and AgNO ₃	2%	2%	0%

Reaction conditions: the mixture of **1a** (0.2 mmol), AgNO₃ (0.04 mmol), K₂S₂O₈ (0.4 mmol), **2** (0.3 mmol) in acetone/H₂O (v/v = 1/1, 1.5 mL) was stirred at 80 °C under N₂ for 12 h. The conversion and yield were determined by ¹H NMR using mesitylene as an internal standard.

Figure R-3

DFT calculations are included (Figure 4) that are consistent with a lower energy pathway for 1,2-silyl transfer relative to 1,5-HAT. The manuscript would be improved by inclusion of citations to support the choice of this basis set and functional, and explicit acknowledgement of the limitations of these DFT calculations (c.f. Chem. Sci. 2020, 11, 217 & references therein). The documentation for these DFT calculations in the supporting information is inadequate for their evaluation, and must be improved.

Response: Thanks for the reviewer's comment. (1) In this work, all geometrical structures were optimized in solvent (acetonitrile) phase with M06-2X/def2-TZVP considering the balance of efficiency and data accuracy. According to the reviewer's suggestion, we have done the single point calculation by various methods to test our computational methods. As shown in Table S2, the calculated solvation single-point energies showed small differences between each other. Therefore, the reported energy data is reliable. Considering the calculating efficiency and accuracy, M06-2X is suitable for geometrical structures optimization. In fact, M06-2X is a widely used method for calculation (*J. Am. Chem. Soc.* **2017**, *139*, 1726-1729; *J. Am. Chem. Soc.* **2016**, *138*, 8247-8252; *Angew. Chem. Int. Ed.* **2018**, *57*, 13795-13799.; *Angew. Chem. Int. Ed.* **2019**, *58*, 16252-16259.; *Chem. Sci.*, **2016**, *7*, 6407-6412.). The energy profiles calculated with M06-2X/def2-TZVP were given in the manuscript and the energy profiles calculated with other methods (Table S2) were involved in the revised supplementary information.

Table S2. Single Point Calculation at the Pathways for 1,2-SiT vs 1,5-H AT (various methods)

Entry	$\Delta G(\text{solv})$ by M06-2X/ def2-TZVP	$\Delta G(\text{solv})$ by M06-2X-D3/ def2-TZVP	$\Delta G(\text{solv})$ by ω B97XD/ def2-TZVP	$\Delta G(\text{solv})$ by B3-PW91/ def2-TZVP	$\Delta G(\text{solv})$ by PBE/ def2-TZVP
A	0.0	0.0	0.0	0.0	0.0
B-ts	1.3	1.3	2.3	2.0	1.8
C-ts	14.3	14.1	14.1	12.2	12.1
D	-26.6	-26.6	-22.5	-21.5	-21.6
E	4.9	4.8	6.7	7.7	7.1

According to the reviewer's comment, we have done the single point calculation by using some other basis set to test the current one. As shown in Table S3, the calculated solvation single-point energies showed small differences with the selected basis sets. Considering the calculating efficiency and accuracy, def2-TZVP is suitable basis set for geometrical structures optimization. Similarly, def2-TZVP is a widely used basis set for theoretical calculations in organocatalysis (*J. Am. Chem. Soc.* **2020**, *142*, 18213-18222.; *J. Chem. Theory Comput.* **2011**, *7*, 2766-2779.). The energy profiles calculated with various basis set (Table S3) were added in the revised supplementary information.

Table S3. Single Point Calculation at the Pathways for 1,2-SiT vs 1,5-H AT (various basis set)

Entry	$\Delta G(\text{solv})$ by M06-2X/ def2-TZVP	$\Delta G(\text{solv})$ by M06-2X/ def2-SVP	$\Delta G(\text{solv})$ by M06-2X/ 6-311+g(d,p)	$\Delta G(\text{solv})$ by M06-2X/ 6-31+g(d,p)	$\Delta G(\text{solv})$ by M06-2X/ def2-QZVPP
A	0.0	0.0	0.0	0.0	0.0
B-ts	1.3	2.1	3.0	2.4	1.2
C-ts	14.3	12.6	14.0	13.7	14.4
D	-26.6	-24.9	-22.3	-22.5	-26.6
E	4.9	3.8	4.2	5.0	4.7

(2) Based on the comments of the reviewers, we cited this references in the revise manuscript. In that work, the energy 1,7-HAT process is 3.73 kcal/mol higher than 1,6-HAT. Meanwhile, the energy difference between 1,5-HAT and 1,2-SiT is much higher (13 kcal/mol) in our work. Moreover, according to these benchmark data (Table S2 and Table S3) as well as considering the efficiency of our computational results, we believe that the calculation method we used here, namely the solvation calculation in M06-2X based on def2-TZVP basis set, would be suitable to afford reasonable explanation in the chemoselectivity of 1,2-SiT vs 1,5-HAT.

(3) We added computational section in the revised revised supplementary information (SI 3.i-k) as follows:

i) Computational methods

All of the DFT calculations were performed with the Gaussian 09 series of programs.^[1] The M06-2X^[2] functional and standard def2-TZVP^[3] basis set were used for geometry optimization in the solvent phase. Harmonic vibrational frequency calculations were performed for all of the stationary points to confirm whether they are local minima or transition structures and to derive the enthalpies and thermochemical corrections for the free energies.

[1]. Frisch, M. J. Gaussian 09, Revision D.01, Gaussian, Inc., Wallingford, CT, (2013). The full author list is shown in the ESI 3.j).

[2]. Zhao, Y. & Truhlar, D. G. The M06 suite of density functionals for main group thermochemistry, thermochemical kinetics, noncovalent interactions, excited states, and transition elements: two new functionals and systematic testing of four M06-class functionals and 12 other functionals, *Theor. Chem. Account.*, **120**, 215 (2007).

[3]. Hellweg, A., Hättig, C. & Höfener, S. et al. Optimized accurate auxiliary basis sets for RI-MP2 and RI-CC2 calculations for the atoms Rb to Rn, *Theoretical Chemistry Accounts.* **117**: 587–597 (2007).

j) Complete reference for Gaussian 09

Frisch, M. J.; Trucks, G. W.; Schlegel, H. B.; Scuseria, G. E.; Robb, M. A.; Cheeseman, J. R.; Scalmani, G.; Barone, V.; Mennucci, B.; Petersson, G. A.; Nakatsuji, H.; Caricato, M.; Li, X.; Hratchian, H. P.;

Izmaylov, A. F.; Bloino, J.; Zheng, G.; Sonnenberg, J. L.; Hada, M.; Ehara, M.; Toyota, K.; Fukuda, R.; Hasegawa, J.; Ishida, M.; Nakajima, T.; Honda, Y.; Kitao, O.; Nakai, H.; Vreven, T.; Montgomery, Jr., J. A.; Peralta, J. E.; Ogliaro, F.; Bearpark, M.; Heyd, J. J.; Brothers, E.; Kudin, K. N.; Staroverov, V. N.; Keith, T.; Kobayashi, R.; Normand, J.; Raghavachari, K.; Rendell, A.; Burant, J. C.; Iyengar, S. S.; Tomasi, J.; Cossi, M.; Rega, N.; Millam, J. M.; Klene, M.; Knox, J. E.; Cross, J. B.; Bakken, V.; Adamo, C.; Jaramillo, J.; Gomperts, R.; Stratmann, R. E.; Yazyev, O.; Austin, A. J.; Cammi, R.; Pomelli, C.; Ochterski, J. W.; Martin, R. L.; Morokuma, K.; Zakrzewski, V. G.; Voth, G. A.; Salvador, P.; Dannenberg, J. J.; Dapprich, S.; Daniels, A. D.; Farkas, O.; Foresman, J. B.; Ortiz, J. V.; Cioslowski, J.; and Fox, D. J. Gaussian 09, revision D.01; Gaussian, Inc.: Wallingford, CT, **2013**.

k) M06-2X/def2-TZVP calculated absolute energies, enthalpies, and free energies of all structures

Table S1. M06-2X/def2-TZVP calculated absolute energies, enthalpies, and free energies of all structures

Geometry	$E_{(\text{elec-M06-2X})}^1$	$H_{(\text{corr-M06-2X})}^2$	$G_{(\text{corr-M06-2X})}^3$	IF^4
A	-951.296797	0.387945	0.314753	-
B-ts	-951.293482	0.386801	0.313463	-244.78
C-ts	-951.266657	0.381218	0.307324	-1642.74
D	-951.338050	0.387487	0.313685	-
E	-951.287505	0.386754	0.313282	-

¹The electronic energy calculated by M06-2X in acetonitrile solvent. ²The thermal correction to enthalpy calculated by M06-2X in acetonitrile solvent. ³The thermal correction to Gibbs free energy calculated by M06-2X in acetonitrile solvent. ⁴The M06-2X calculated imaginary frequencies for the transition states.

The proposed mechanism (Figure 5) is plausible, and citations explicitly related to the roles of oxidants need to be included in the context of the associated discussion either in the manuscript or supporting information.

Response: Thanks for the comments. $K_2S_2O_8$ has been reported to be able to oxidize Ag(I) to Ag(II). The related papers (*J. Am. Chem. Soc.* **2017**, *139*, 12430–12433 and *J. Am. Chem. Soc.* **2015**, *137*, 9820–9823) have been added as reference 52 and 53 in the revised manuscript.

With a cursory scan of Figures 6 and 7, the grey circles on top of some carbon atoms appear similar to oxygen atoms, so it may be worthwhile to reconsider whether their use in this document is beneficial.

Response: Thanks for the comments. We modified the grey circles in the revised manuscript.

Supporting information (SI):

1. Many procedures appear well drafted.
2. Many ^{13}C NMR spectra in the supporting information do not meet the ACS definition for characterization-quality spectra and do not provide adequate evidence that the compounds can be isolated in high purity (c.f. pages S54, S58, S62, S70, etc).

Response: Thanks for the comments. We double checked the spectra and provided qualified ^{13}C NMR spectra in the revised supplementary information.

Reviewer #3 (Remarks to the Author):

In this manuscript, Yang, et. al. introduces a silyl group at α -position of alcohol that can inhibit 1,5-hydrogen atom transfer (1,5-HAT) and make various α -functionalization products from alcohol substrate. Notably, this method employed silver catalyst and use cheap oxidant. Author also bring a

novelty to selectively transform alcohol through 1,2-silyl transfer (1,2-SiT) instead of 1,5-HAT. The significance of this paper for the targeted journal is good. This manuscript provides broad substrate scopes, several mechanistic studies and computational calculation to support the plausible mechanism. However, scopes of alcohol were limited to secondary alcohol. The manuscript may be publishable once the following items have been addressed. My detailed analysis of this manuscript are as follow

Specific comments

1. In influence of silyl groups on the efficiency of the reaction (Section 1.2), is the different substrate of silyl group affect C-Si bond length and BDE? I think C-Si bond and BDE is also important factor in 1,2SiT.

Response: Thanks for the kind suggestion. We revised the manuscript by adding the following statement: “Moreover, the different substituents of the silyl groups also affect the C-Si bond length and bond dissociation energy, which might also be important factors in 1,2-SiT.”

2. In mechanism study (Fig. 3c), “Protected α -silyl alcohol 1a-2 only gave 5% yield of compound 3a-2 (Fig. 3c), indicating that the generation of carbon radical via direct oxidative cleavage of C-Si bond is less likely to be the major pathway in the reaction with 1a.” These results show that a direct oxidation mechanism of α -silyl alcohol is also possible. In general, when a silyl group is introduced at the α -position of an alcohol, the oxidation potential is reduced compared to the free alcohol. But α -silyl alcohol still has high oxidation potential Therefore, a strong oxidant is necessary. In order to exclude the direct oxidation mechanism, it is recommended to add information on the oxidation potential of alpha silyl alcohol and related reactions to the reference. *J. Am. Chem. Soc.* 1990, 112, 1962-1970. *Angew. Chem. Int. Ed.* 1998, 37, 660-662. *Org. Lett.* 2017, 19, 4696-4699. *Org. Lett.* 2018, 20, 6239-6243. *Chem. Commun.* 2020, 56, 2873-2876.

Response: Thanks for the kind suggestions. Oxidation potentials of alpha silyl alcohol **A** and protected α -silyl alcohol **B** are similar (**Figure R-4**, *J. Am. Chem. Soc.* 1990, 112, 1962). We added the related papers as references 42~46 in the revised manuscript. The manuscript was revises as follows: “Protected α -silyl alcohol **1a-2** only gave 5% yield of compound **3a-2** (Fig. 3c), indicating that the generation of carbon radical via direct oxidative cleavage of C-Si bond is less likely to be the major pathway in the reaction with **1a**.⁴²⁻⁴⁵ This result was inconsistent to the similar oxidation potential of α -silyl alcohol and the methyl protected counterpart.⁴⁶”

Ref: *J. Am. Chem. Soc.* **1990**, 112, 1962

Figure R-4

3. In mechanistic study (Fig.3e), the role of water was examined with a reaction of standard condition without water as a solvent. In this case, why triphenyl silyl substituted alcohol 1a-4 used instead of phenyl dimethyl silyl substituted alcohol? Meanwhile another mechanistic study used phenyl dimethyl silyl substituted alcohol.

Response: Thanks for the comments. We tried the phenyl dimethyl silyl substituted alcohol, but less than

2% yield of silyl ether was isolated due to the instability of O-SiMe₂Ph bond under the reaction conditions.

4. In alcohol scopes, there were many secondary alcohol scopes but just only one example of tertiary alcohol. Do tertiary alcohol substrates fail under the report reaction conditions or were they not tried? I think it would be of interest to the readership for this information to be provided.

Response: Thanks for the comments. Tertiary alcohols work, but in lower yield than secondary alcohols. Further optimization of the reaction conditions resulted in 41% isolated yield of **3ab** (Figure R-5, entry 5). Two more examples **3aa** and **3ab** have been added into the revised manuscript.

Figure R-5

5. In the Menisci type reaction, quinoline type of substrates were studied, the more general heterocycles weren't considered. It is recommended to add reactions of various heterocyclic compounds.

Response: Thanks for the comments. Quinolines and isoquinolines were suitable substrates. We tried many pyridines, such as **B1**, **B2**, **B3** and **B4**, but less than 5% yield were afforded (Figure R-1). Other heteroarenes such as **B5**, **B6**, **B7**, **B8** and **B9** were also not successful substrates under the current conditions. A yield of 9% were obtained when **B10** was employed. Further optimization of the reaction conditions with **B10** as the substrate resulted in the product in 35% isolated yield of product **C10** (Figure R-2). We added the data of **C10** in the revised manuscript. The failed substrates were added in the revised supplementary information.

The yield was determined by ^1H NMR using Mesitylene as an internal standard.

Figure R-1

Entry	Solvent	Yield. of C10
1	Acetone:H ₂ O (5:1)	5
2	DMSO:H ₂ O (5:1)	5
3	DMSO:H ₂ O (5:2)	19
4	DMSO:H ₂ O (5:3)	25(35 ^a)
5	DMSO:H ₂ O (5:4)	19
6	DMSO:H ₂ O (5:5)	8

Yield of **C10** were determined by ^1H NMR using Mesitylene as an internal standard. ^a isolated yield

Figure R-2

6. In this manuscript there were no gram scale reaction and further application (i.e. transformation further of product) to check the valuable of the product.

Response: Thanks for the kind suggestions. Gram scale reaction afforded compound **3a** in 60% yield (Figure R-6a). The product **3a** could be transformed to compound **1a-2** in 57% yield, and **3a-8** in 72% yield (Figure R-6b). We added these information in the revised manuscript.

Figure R-6

Minor Comments

1. Fig. 4 caption: “Free-energy profile of the pathways for 1,2-SiT vs 1,5-H AT...”
1,5-HAT sentences should not contain space between H and AT

Response: Thanks for the kind suggestions. We corrected the typo in the revised manuscript.

2. In SI page S5, compound 1h and 3h are same. Please fix it.

Response: Thanks for the kind suggestions. We corrected the supplementary information.

3. In SI NMR spectra, the quality were good but some of spectra still contain impurity or solvent junk. I think the addition of title in each spectrum would be better (i.e. ¹H NMR spectrum, ¹³C NMR spectrum, or ¹⁹F NMR spectrum) and also insets with enlarged multiplets should be included on spectra.

Response: Thanks for the kind suggestions. We revised the supplementary information accordingly.

REVIEWERS' COMMENTS

Reviewer #2 (Remarks to the Author):

The authors disclose two practical advances that rely on a conceptual insight – the recognition that, once an alkoxy radical is generated, 1,2-silyl transfer would be more kinetically feasible than 1,5-hydrogen atom transfer (1,5-HAT), and the resultant α -alkoxy radical could be trapped by known trapping agents such as an oxime ether (c.f. Table 1, Figure 6), or a heteroarene (Figure 7). This research could merit publication in *Nature Commun.* with minor revisions.

Experiments with TEMPO should be moved to the supporting information, and not discussed in the main body of the manuscript. The control experiments are useful. Nevertheless, as noted before “Even with these control reactions, an experiment involving TEMPO would likely prove inconclusive.”

The proposed mechanism (Figure 5) is plausible, and citations have been added (i.e. ref 52, 53); however, it would seem more appropriate to cite Walling and Camaioni, Role of silver(II) in silver-catalyzed oxidations by peroxydisulfate. *J. Org. Chem.* 1978, 43, 3266, which documents the oxidation of Ag(I) by peroxydisulfate anion, and subsequent Ag-O bond-homolysis to generate an alkoxy radical. By contrast, references 52 and 53 don't provide any support for this mechanism, and it would be appropriate to remove them from the manuscript.

In relation to DFT calculations, a number of citations are given to other manuscripts with M06-2X calculations using related systems. In this context, reference 48 does not appear to include M06-2X calculations, and reference 49 appears to evaluate polar intermediates and transition states (rather than radical intermediates and transition states). So these references do not appear particularly relevant to the sentence at which they are cited.

Reviewer #3 (Remarks to the Author):

The authors have addressed the suggestions of the previous reviewers satisfactorily. All in all, I recommend publication of this work as it is.

RESPONSE TO REVIEWERS' COMMENTS

Reviewer #2 (Remarks to the Author):

The authors disclose two practical advances that rely on a conceptual insight – the recognition that, once an alkoxy radical is generated, 1,2-silyl transfer would be more kinetically feasible than 1,5-hydrogen atom transfer (1,5-HAT), and the resultant α -alkoxy radical could be trapped by known trapping agents such as an oxime ether (c.f. Table 1, Figure 6), or a heteroarene (Figure 7). This research could merit publication in Nature Commun. with minor revisions.

Response: Thanks for the comments.

Experiments with TEMPO should be moved to the supporting information, and not discussed in the main body of the manuscript. The control experiments are useful. Nevertheless, as noted before “Even with these control reactions, an experiment involving TEMPO would likely prove inconclusive.”

Response: Thanks for the suggestions. The experiments with TEMPO have been moved to the supporting information.

The proposed mechanism (Figure 5) is plausible, and citations have been added (i.e. ref 52, 53); however, it would seem more appropriate to cite Walling and Camaioni, Role of silver(II) in silver-catalyzed oxidations by peroxydisulfate. J. Org. Chem. 1978, 43, 3266, which documents the oxidation of Ag(I) by peroxydisulfate anion, and subsequent Ag-O bond-homolysis to generate an alkoxy radical. By contrast, references 52 and 53 don't provide any support for this mechanism, and it would be appropriate to remove them from the manuscript.

Response: Thanks for the suggestions. We removed the reference 52, 53 and cited J. Org. Chem. 1978, 43, 3266 (ref 50 of the revised manuscript).

In relation to DFT calculations, a number of citations are given to other manuscripts with M06-2X calculations using related systems. In this context, reference 48 does not appear to include M06-2X calculations, and reference 49 appears to evaluate polar intermediates and transition states (rather than radical intermediates and transition states). So these references do not appear particularly relevant to the sentence at which they are cited.

Response: Thanks for the suggestions. We removed the reference 48 and 49.

Reviewer #3 (Remarks to the Author):

The authors have addressed the suggestions of the previous reviewers satisfactorily. All in all, I recommend publication of this work as it is.

Response: Thanks for the comments.